# Latent Diffusion Models with Image-Derived Annotations for Enhanced AI-Assisted Cancer Diagnosis in Histopathology

**DOI:** 10.3390/diagnostics14131442

**Published:** 2024-07-05

**Authors:** Pedro Osorio, Guillermo Jimenez-Perez, Javier Montalt-Tordera, Jens Hooge, Guillem Duran-Ballester, Shivam Singh, Moritz Radbruch, Ute Bach, Sabrina Schroeder, Krystyna Siudak, Julia Vienenkoetter, Bettina Lawrenz, Sadegh Mohammadi

**Affiliations:** 1Decision Science & Advanced Analytics, Bayer AG, 13353 Berlin, Germany; guillermo.jimenezperez@bayer.com (G.J.-P.); javier.montalttordera@bayer.com (J.M.-T.); jens.hooge@bayer.com (J.H.); guillem.duranballester@bayer.com (G.D.-B.); shivam.singh2@bayer.com (S.S.); sadegh.mohammadi@bayer.com (S.M.); 2Pathology and Clinical Pathology, Bayer AG, 13353 Berlin, Germany; moritz.radbruch@bayer.com (M.R.); ute.bach@bayer.com (U.B.); sabrina.schroeder@bayer.com (S.S.); krystyna.siudak@bayer.com (K.S.); julia.vienenkoetter@bayer.com (J.V.); bettina.lawrenz@bayer.com (B.L.)

**Keywords:** synthetic data, histopathology, generative AI, diffusion models, cancer detection, AI in health care, digital health, medical imaging

## Abstract

Artificial Intelligence (AI)-based image analysis has immense potential to support diagnostic histopathology, including cancer diagnostics. However, developing supervised AI methods requires large-scale annotated datasets. A potentially powerful solution is to augment training data with synthetic data. Latent diffusion models, which can generate high-quality, diverse synthetic images, are promising. However, the most common implementations rely on detailed textual descriptions, which are not generally available in this domain. This work proposes a method that constructs structured textual prompts from automatically extracted image features. We experiment with the PCam dataset, composed of tissue patches only loosely annotated as healthy or cancerous. We show that including image-derived features in the prompt, as opposed to only healthy and cancerous labels, improves the Fréchet Inception Distance (FID) by 88.6. We also show that pathologists find it challenging to detect synthetic images, with a median sensitivity/specificity of 0.55/0.55. Finally, we show that synthetic data effectively train AI models.

## 1. Introduction

Histopathology is considered the gold standard for cancer diagnostics. It entails the microscopic examination of tissue samples to identify signs of disease. These tissue samples, usually obtained through surgical resections or biopsies, are prepared and examined using hematoxylin and eosin (H&E) staining protocols. Subsequently, they can be digitized into gigapixel-sized whole-slide images (WSIs). H&E imaging offers insights into the structural and morphological changes associated with a variety of pathological conditions, including cancer. When presented with histopathological images, computer-aided diagnosis (CAD) systems have shown significant potential in classifying diseases, detecting genetic alterations, and quantifying the size and presence of lesions [1,2,3,4,5,6].

**Challenges**. However, the use of this technology in the medical field poses significant challenges: firstly in the handling of large WSIs and the high variability of staining and slide preparation techniques and secondly, the acquisition of sufficiently large datasets for model training is hindered by the rarity of diseases, high acquisition costs, and reliance on low-availability technologies such as next-generation sequencing [3,5]. The lack of training data in the medical domain often results in overconfident predictions from highly complex AI models, which may not apply to real-world data. Although in-domain data augmentation techniques such as flipping, cropping, or stain normalization address some of these issues [5,7], they inadequately represent the variability of a real-world dataset. Consequently, these augmentations only marginally improve model generalization when addressing bias and data imbalances [8].

**Background**. In response to these challenges, generative models have been used to tackle data scarcity, imbalance, and privacy protection. Notably, they enable privacy-preserving model training by generating images that are not directly derived from real subjects [9,10,11,12]. Certain models, such as generative adversarial networks (GANs), have demonstrated significant potential in producing diverse and high-fidelity images [13,14,15,16], yet they often encounter mode collapse and training instability when hyperparameters are not meticulously adjusted [17]. The more recent introduction of latent diffusion models (LDMs) has gained interest for their ability to produce high-resolution images from text descriptions. LDMs function through a *forward process* and a *reverse process*. The former gradually adds Gaussian noise to a data sample’s latent representations, while the latter, signifying data synthesis, subjects the latent representation to iterative denoising steps, typically carried out by a neural network. They since have been successfully applied to a wide range of applications, such as image synthesis [18,19,20,21,22,23], image super-resolution [24], image-to-image translation [25], image editing [26], inpainting [27], video synthesis [28,29], and text-to-3D generation [30,31]. There is a growing number of studies on histopathological image synthesis demonstrating that LDMs can produce diverse and high-quality images similar to GANs, without the inherent instabilities of adversarial model training [32,33,34,35]. Aversa et al. introduced an LDM-based sampling method that uses proprietary annotated cellular macro-structure data from WSIs [33]. Their method is capable of producing arbitrarily large histopathological images, taking an important step towards WSI generation. However, their reliance on pixel-wise annotations consists of a limitation, since in histopathology, their creation involves non-routine, time-consuming, and costly manual labeling procedures. In contrast, Yellapragada et al. [34] conditioned an LDM on ChatGPT summaries of WSI-level pathological reports and classifier-driven annotations at the patch level, showing good generation metrics on the TCGA-BRCA dataset. Nevertheless, WSI-level reports do not necessarily include descriptive information for all the regions of the WSI, thus making it a non-ideal candidate for conditioning the generation of WSI patches and possibly limiting image quality. In addition, these WSI reports are also not always available. Finally, Ye et al. [35] further minimized the annotation requirements by firstly training an unconditional diffusion model (i.e., trained without any annotations) for then fine-tuning it with a smaller annotated dataset. Their approach, however, relies on the assumption of close similarity between pretraining and target data distributions. Such an extensive and diverse dataset can be costly to produce, potentially restricting the applicability of their approach.

**Proposal**. This work presents the **M**orphology-**E**nriched **P**rompt-building (MEP) approach: a data-centric methodology to extract morphological text descriptors from H&E image patches, which can be used as additional conditioning variables for training histopathology text-to-image LDMs. It conceptually improves upon existing methods by allowing histopathology image generation without extensively annotated data neither at the pixel nor WSI level. Additionally, it does so without relying on large-scale pretraining with a larger corpus of data. MEP-based LDM fine-tuning improves image generation as compared to (i) not fine-tuning LDM models or (ii) fine-tuning using only binary labels as conditioning variables (see Figure 1).

For this purpose, Patch Camelyon (PCam) [36,37,38], a dataset composed of 96 × 96 px patches extracted from WSIs of lymph node sections, was selected. Each patch is loosely annotated with a binary label indicating the presence or absence of metastatic tissue, and no WSI-level reports are provided. Then, a pretrained image encoder model (DiNO [39]) was leveraged to capture rich cellular configurations, overarching tissue structures, and distinctive pathological landmarks in the image patches, independently of the exhaustiveness of the dataset’s annotations. The outputs of the DiNO model were postprocessed using a k-means clustering algorithm to agglomerate similar data points based on feature similarity, facilitating the generation of informative prompts/captions (e.g., *“Histology image of **healthy** tissue, morphology type **five**”*). These prompts are in turn utilized as richer inputs for fine-tuning a pretrained LDM model (the open-source Stable Diffusion [21] model pretrained on natural images [40]), thereby better representing the data distribution captured by the model.

Assessing the realism and utility of synthetic images is a difficult endeavor that is highly debated in the scientific literature [41,42], so multiple evaluation settings were defined. Firstly, image quality is assessed qualitatively by visual inspection of the real and synthetic datasets as well as corresponding feature distributions. The quantitative evaluation relied on standard metrics assessing sample quality and coverage of the data distribution, namely the Fréchet Inception Distance (FID) [43] and the improved precision and recall [44]. The domain experts’ ability to discern real from synthetic images was assessed via blind read as a supplementary qualitative image quality assessment for the proposed method. Finally, synthetic image quality was indirectly assessed via two distinct cancer-detection subtasks. On the one hand, a classifier was trained using synthetic images only, to test how well the generative model replicates the discriminative features between cancerous and healthy tissue. On the other, to further explore the relevance of the synthetic data to the cancer-detection task, multiple classifier models were trained with varying amounts of real and synthetic data. As a secondary outcome, this experiment explores the applicability of our generated data for mitigating data acquisition costs and privacy concerns. Figure 2 delineates the main elements of the proposed methodology.

**Contributions.** *(i)* We analyze the capabilities of a natural imaging pretrained LDM for histopathology image generation before and after fine-tuning with scarcely annotated in-domain data; *(ii)* we devise MEP, a novel prompt-building strategy to improve synthetic image quality in annotation-scarce scenarios; *(iii)* we qualitatively and quantitatively compare it to a binary class conditional generation baseline; and *(iv)* we test the downstream utility of the proposed method to mitigate privacy concerns and data acquisition costs.

## 2. Materials and Methods

### 2.1. Data and Preprocessing

The dataset used in this work is Patch Camelyon (PCam) [36,37,38], which is a common image classification benchmark consisting of 327,680 patches of 96 × 96 px in size, extracted from 400 histopathologic scans of sentinel lymph node sections from breast cancer patients. The version of this dataset that we used is the curated one from Kaggle [38], which removes all duplicated patches and comes with a default train/test split. The train split contains 220,025 labeled images, where no annotation is included other than a single binary class label per patch indicating the presence or absence of metastatic tissue in the 32 × 32 center region. The official test comprises 57,486 images for which no labels are provided, therefore all the reported classifier performance evaluations required the submission of the predictions to Kaggle. The predominant cell morphology types represented in the dataset are metastatic cancer cells, lymphocytes, and stroma.

The dataset was further preprocessed to remove image patches lacking meaningful tissue characteristics that we want to be able to synthesize, i.e., no-foreground patches. This step yielded a clean dataset of 216,868 image patches. See more details about the dataset and the outlier removal step in Section A.1.

### 2.2. Generative Architecture

The image generator architecture is a vanilla LDM model, fine-tuned from a specific pretrained model by the name of Stable Diffusion (SD) [21]. It comprises 2 main components: a *variational autoencoder* (VAE) and a *conditional denoising UNet*.

The VAE is a stand-alone model that is pretrained to convert a high-dimensional input into a lower-dimensional representation from which the original data can be retrieved with minimal information loss. In the context of LDMs, the encoding branch of the VAE (E) is used to encode the training images $x\in R^{H\times W\times 3}$ onto a lower dimensional latent space $z=E\left(x\right)\in R^{h\times w\times c}$ (with h<H and w<W), upon which the training and generation process takes place. Here, *H* and *W* are the height and the width of the input images, and 3 represents the three color channels present in a typical WSI, while *h*, *w*, and *c* are the height, width, and number of channels of the latent *z*, respectively. Latents in this space can be decoded back to the original image space via the VAE’s decoder branch $\hat{x}=D\left(z\right)$.

In turn, the backbone of the LDM is a *conditional denoising UNet* whose function is to model the VAE’s latent space. In particular, this network is trained to minimize a denoising objective in that space via an iterative approach while incorporating in that process optional conditional information. This conditional information can range from class labels, segmentation masks, to text, which is the one used in our pipeline. For the case of text, a CLIP model [45] is used to convert the text into rich embedded representations that can be introduced in the denoising process through a cross-attention mechanism.

Considering all these components, a training step of an LDM with text conditioning can be separated into two distinct processes, the *forward* and *reverse diffusion* steps. In the *forward* process, given an image from the training data distribution ($x_{0}$), we compute its representation in the latent space of the VAE ($z_{0}=E\left(x_{0}\right)$) as well as the CLIP text embedding of its corresponding text prompt (c(y)). A time step is sampled from a uniform distribution (t∼U(1,…,T)) and random noisy latent is sampled from a normal distribution (ϵ∼N(0,1)). A corrupted latent ($z_{t}$) is created by combining ϵ with $z_{0}$ via a scheduler that depends on the sampled *t*, ensuring that the larger the *t*, the more noise is added to $z_{0}$. In the *reverse* diffusion process, a UNet is used to predict the initially sampled noise, ϵ, based on the sampled *t*, $z_{t}$, and its corresponding text embedding c(y).

The MSE loss between the predicted and true noises allows for the computation of a gradient that can be used to update the weights of the text encoder (CLIP) and, most importantly, of the UNet. See Equation (Equation 1):(1)$$L_{LDM}=E_{z∼\epsilon (x),\;y,\;\epsilon ∼N(0,1),\;t}\left[{||\epsilon -\epsilon_{\theta }\left(z_{t},\;t,\;c\left(y\right)\right)\left|\right|}_{2}^{2}\right]$$

For image synthesis, a random noisy latent is sampled ($z_{T}∼N\left(0,1\right)$), and using the user-provided conditioning text (*y*), it is possible to recursively denoise $z_{T}$ into a new uncorrupted latent ($z_{0}$). A new image can then be reconstructed with the decoding branch of the VAE, generating a new image $x=D(z_{0})$ which matches the description provided by *y*. In our case, given SD’s particular pretraining, the synthetic images are generated at a resolution of 512 × 512.

### 2.3. Prompt Building

Two prompt-building approaches for SD fine-tuning are explored in this work. While the baseline approach leverages solely the provided patch labels for binary class-conditional image synthesis, our morphology-enriched approach further extracts semantic information from the images for more varied and realistic image generation.

#### 2.3.1. Class-Based Prompt Building

As the baseline approach, a text prompt is created for each of the patches in the curated dataset by filling a template with the name of the corresponding class label: *“Histology image of $\langle \mathrm{LABEL}\rangle$ tissue”*, where LABEL corresponds to either “cancer” or “healthy”.

#### 2.3.2. Morphology-Enriched Prompt Building (MEP)

The proposed morphology-enriched prompt-building approach (MEP) is a three-step process: feature extraction, k-means clustering, and prompt template completion.

**Feature Extraction.** Image features were extracted from each image in the dataset by a pretrained DINO-ViT [39], which has been shown to encode useful semantic information across tasks and domains [39,46]. The base ViT variant with 8 × 8 patches (B/8) was employed, which had the best linear classification performance in ImageNet [39]. The output feature vector was set to be equal to the class token, a 768-dimensional vector that aggregates information from the entire image. Further details can be found in the supplementary Section A.2.

**K-Means Clustering.** In order to determine morphological subgroups, we applied k-means clustering on extracted feature vectors. We determined the optimal number of clusters (*k* = *33*), based on the smallest SD-index [47] for *2 <= k <= 50*, as it indicates a good balance between compactness and separation of clusters.

**Prompt Template Completion.** For each image, a novel template is filled with both the existing patch labels and extracted cluster information for each image: *“Histology image of $\langle \mathrm{LABEL}\rangle$ tissue, morphology type $\langle \mathrm{INDEX}\rangle$”*, where **INDEX** is the k-means cluster index to which a given image belongs and **LABEL** corresponds to either “cancer” or “healthy”. Following this method, 66 unique text prompts (based on 2 labels and 33 clusters) were generated for the 216,868 image dataset which, compared to the *baseline approach*, encode for a much wider variety of tissue morphology and staining profiles, complementing the existing label information.

### 2.4. Dataset Balancing

Due to the diversity and heterogeneity of histopathology data, it is likely that the distribution of the different tissue morphology and staining profiles is not uniform across the curated PCam dataset. To prevent this unbalanced distribution from introducing bias during both the SD’s fine-tuning or inference towards specific image subtypes, we resampled the dataset to obtain a distribution of examples across prompts and classes that was as close as possible to uniform. The top 21 most populated prompts from each class (42 in total) were selected and undersampled to create a subset of 51,000 examples where every prompt and class label is approximately equally represented (either 1214 or 1215 examples per prompt). This subset is then split into a 50,000 training sample subset and 1000 validation sample subset, also ensuring the same balanced prompt distribution is kept in both sets. The choice of the final curated balanced dataset’s size determines the number of prompts that can be selected. The 50,000 examples are used to minimize computational effort during the SD’s fine-tuning while also being large enough to provide statistical robustness for the downstream image quality metric computations. The 1000 validation samples are used as an independent holdout set used to monitor the performance of the downstream classifier models during training so as to determine the best model configuration that should be used for testing, which is standard practice in Deep Learning.

Moreover, despite this curated subset having been selected based on the morphology-enriched prompt set, the same exact 50,000 training images are used for the baseline approach so as to ensure comparability between the two methods. This is achieved by duplicating the selected dataset and simply removing the extra morphology information from the prompts (i.e., *“(…), morphology type $\langle \mathrm{INDEX}\rangle$”*). Given the nature of both prompt-building approaches, this will also ensure that the balanced distribution across prompts and labels is maintained in the dataset captioned via the baseline approach.

### 2.5. Fine-Tuning and Image Synthesis

The same 50k curated dataset from PCam is prompted via the two different approaches explored in this work, and then each set of prompts is used independently to fine-tune SD, yielding two different SD models. Both fine-tuning experiments were run for 12.5 k training steps, using a float16 precision (fp16), a learning rate of $10^{-5}$, and an effective batch size of 64 (4 GPUs with a batch size of 8 and gradient accumulation of 2). The GPUs used were NVIDIA A10G Tensor Core GPUs and took approximately 12 h for 12.5 k training steps.

The SD checkpoint version 1.5 was used (*runwayml/stable-diffusion-v1-5* [21]) and the UNet’s, the VAE’s, and the CLIP’s text encoder’s weights were kept frozen during fine-tuning. Each of the resulting fine-tuned LDMs is then used for inference with the corresponding prompts, yielding two synthetic datasets of 50 k images each also balanced on prompts and class, mirroring the corresponding real one, but at an increased resolution of 512 × 512 px. Further details on the implementation of the training and image synthesis pipeline can be found in the supplementary Section A.2.

### 2.6. Evaluation Protocol

#### 2.6.1. Quantitative Image Quality Assessment

The quality of the synthetic images was quantitatively measured via the Fréchet Inception Distance (FID) [43], which is the standard evaluation metric for assessing how close the distribution of real and synthetic data are (image fidelity). Rather than comparing the images pixel by pixel, it is based on the mean and standard deviation across each dataset of the corresponding Inception-V3 features (1 × 1028) [48], a convolutional neural network pretrained on a large natural image dataset (ImageNet). The precision and recall were two additional metrics computed to provide independent assessments of the average sample quality and coverage of the real distribution, respectively [44]. While precision quantifies how much of the synthetic data diversity is realistic, recall quantifies how much of the real data diversity can be found within the synthetic dataset. Like the FID score, these last two metrics were also computed using Inception-V3 features. All metric computations were based on 50k samples from both the real and synthetic datasets, which is standard practice when reporting results of generative AI models. Finally, all metrics were computed using the PyTorch image quality (*piq*) library [49,50].

#### 2.6.2. Qualitative Image Quality Assessment

In turn, the qualitative assessment protocol was two-fold. First, a visual inspection of random subsets of real and synthetic data was conducted. Second, we observe the Inception-V3 embedding distributions on a further reduced 2D feature space. In this space, the overlap between the real and the two synthetic distributions can be visualized, providing us with a visual depiction of how much coverage of the real data distribution each prompt-building approach attains. This dimensionality reduction was conducted via the UMAP technique, which allows one to project Inception-V3 embeddings of both real and synthetic data into a 2D manifold, while preserving as much of the local and more of the global data structure than other methods, like t-SNE [51], with a shorter run time [43,52]. Notably, the 2D UMAP model was fit on only the real data and then used to project every real and synthetic latent to that learned space.

As a supplementary evaluation of the images generated via the proposed MEP method, we designed and executed a Visual Turing Test to assess the plausibility of the generated images as reviewed by five board-certified veterinary pathologists. See Section A.3 for the full description of the experimental setting.

#### 2.6.3. Synthetic-Only Classifier Training

The validation of the proposed MEP approach is also supported by the training of a classifier for distinguishing “cancer” labeled patches from “healthy” ones on each of the following three sets of training data and their subsequent evaluation on the hidden test from Kaggle (see Section 2.1): (a) the 50k real dataset used to train both LDMs, (b) a 50 k synthetic dataset generated via the baseline approach, and (c) a 50k synthetic dataset generated via the proposed MEP method. This was used to evaluate the extent to which synthetic data are able to represent the full distribution of real samples, and also to determine which of the methods better represents the discriminative features between the two classes in the real dataset. The underlying assumption is that the better the generative model captures the true data distribution, the better the classification performance in models trained on the resulting synthetic data [53].

The classifier architecture, hyperparameters, and evaluation setting chosen for this analysis were selected according to Yellapragada et al. [34], and it is fully described in Section A.4 of the Appendix. The best epoch was selected for testing based on its Area Under the Receiver Operating Characteristic Curve (AUC) on the validation subset of 1000 real images described in Section 2.4.

#### 2.6.4. Mixed Real and Synthetic Data Classifier Training

To further assess the extent to which the synthetic data generated via the MEP method are relevant to the cancer-detection task, the evaluation protocol relied on training additional classifier models, although now on multiple proportions of real and synthetic data. Moreover, this experiment will help determine if and when synthetic data can serve as a viable alternative to larger amounts of real data.

Concretely, we conducted experiments of adding synthetic data generated via the proposed method in different proportions and in 7 different training data regimes. Each data regime corresponds to an initial amount of real data and spanned over the following range of values: 10, 25, 50, 100, 500, 1000, and 10,000 real samples. For each data regime, we then added synthetic samples to the training data in 6 different ratios with respect to the initial amount of real data: 0%, 25%, 50%, 100%, 200%, and 300%. These data are sampled in a balanced fashion from the previously mentioned 50 k synthetic dataset generated via our proposed method.

A K-fold cross-validation scheme was used to ensure the robustness of our conclusions to the subsets of real and synthetic data used for training the classifiers. For each initial amount of real data and for each synthetic augmentation ratios, *k* different models (in this case k=10) are trained on distinct subsets of real and synthetic data sampled using 10 different fixed random seeds. In total, 420 different models were trained across all training settings (7 data regimes, 6 ratios, and 10 trainings lead to 420=7×6×10 trained models). The results are then reported as a distribution of the Kaggle test set performances obtained by each of the 10 models for each setting.

The classifier architecture, hyperparameters, and evaluation setting chosen for this analysis is fully described in Section A.4 of the Appendix. The best epoch was selected for testing based on its AUC on the validation subset of 1000 real images described in Section 2.4.

## 3. Results

### 3.1. Histopathology-Specific Stable Diffusion Fine-Tuning

An initial finding was the inability of out-of-the-box SD models to generate histopathology images. In fact, as illustrated in Figure 1b, the images generated by this model without any additional training resemble more artistic interpretations of histopathology images, obviating the need for histopathology-specific fine-tuning.

To address this, an SD model was initially fine-tuned with in-domain medical data and the baseline class-based prompt-building approach (see Section 2.3.1). After fine-tuning, the model was employed to generate class-conditional images from the healthy/cancerous categories. Figure 1c depicts samples of the resulting synthetic dataset.

Notably, the generated images are more realistic and show some level of control over their features (i.e., selectively generating images with or without cancer tissue) as compared to the non-fine-tuned counterpart, underlining the model’s ability to represent complex biomedical concepts when captioned data are available. However, the synthetic images lack variability and seem to represent reduced color and morphological information, which could limit their utility for clinically relevant downstream tasks. To address this, further experiments varying the prompt-building process were performed, with the objective of maximizing information retained by the SD models from a dataset with limited annotations.

### 3.2. Morphology-Enriched Prompt Building

We once again fine-tune the SD model, although this time using the proposed MEP methodology described in Section 2.3.2. First, a qualitative evaluation was conducted through the visual inspection of random subsets of real and synthetic data. The randomly selected subsets from Figure 1 and Figure 3 (top row) depict samples from the real dataset and random subsets of images generated via each prompt-building approach. Regarding coverage of the real data distribution, the morphology-enriched method yields a synthetic dataset containing a much wider variety of images compared to the baseline approach, and which also better approximates the diversity in the real dataset. To further illustrate this, Figure 3 (bottom row) depicts the real and synthetic embedding distributions overlapped in the same plot for each prompt-building approach. As can be seen, training SD using only the label as conditioning tends to produce images from solely two modes, covering significantly less of the real data distribution.

Figure 4a depicts the real and both synthetic distributions overlapped in the same 2D UMAP plot, where multiple manually selected regions are highlighted. Regions R_1_ and R_2_ represent specific subtypes of images from the real distribution that both explored approaches are able to replicate (i.e., the two modes mentioned earlier). In contrast, regions A to F highlight the particular subtypes of images from the real data distribution that only the proposed method is able to synthesize. In these regions, the baseline methods yield only a few artifact-prone images that fail to closely match the features of real images in those regions. In addition, observing the selected examples from each highlighted region on Figure 4, a gradient of increasingly “cancerous”-looking images can be discerned from the left to right side of the UMAP plot.

Additionally, standard image quality metrics were also computed. Table 1 shows that using the MEP approach leads to a significant improvement in image fidelity as quantified by the FID score [43]. The baseline approach from earlier yielded a 178.8 FID score while the morphology-enriched approach achieves an improved score of 90.2. Table 1 shows that the proposed method also improves precision, although at a slight cost of recall.

Furthermore, the results of the supplementary blind evaluation with domain experts shows the distinction between real and synthetic samples generated by the proposed MEP method to be challenging, with a median sensitivity/specificity of 0.55/0.55. The full report on the results including the reader performance, confidence, and agreement can be found in Section B.1.

### 3.3. Synthetic-Only Classifier Training

Next, we trained a classifier on synthetic data and assessed the feature-based discriminative power for “cancer” and “healthy” image patches. Table 2 presents the classification results from this analysis. As expected, training solely on real data still outperformed training on either of the synthetic datasets. However, training on the synthetic dataset generated through our prompt-building approach yields a higher AUC score when compared to the baseline, reducing the gap towards the real data by 0.032 AUC. This can be attributed to the larger reach that the SD model trained on the morphology-enriched prompts has over the image space (see Section 3.2).

### 3.4. Mixed Real and Synthetic Data Classifier Training

Additional classification models were trained on multiple proportions of real and synthetic data generated via our method. Figure 5 aggregates the test set AUC values across 10 cross-validation folds for various settings. As anticipated, the test set AUC exhibits a positive correlation with the quantity of real data used to train the classifier. Additionally, for data regimes with up to 500 real training samples, there is a distinct performance improvement as the amount of added synthetic data increases. Notably, when having only 25 real samples and using an augmentation ratio of 300%, the median performance rises from 0.775 AUC to 0.831 AUC (0.056 increase), surpassing the performance when training on double the amount of real data (0.827 when training on 50 real samples). Also, using an augmentation ratio of 300% when training with 50 real samples leads to a median performance of 0.865 AUC, which is a 0.038 increase from when just using real data (0.827) and is at the level of when training on double the amount of real data, i.e., 100 real samples (0.872 AUC). Similarly, training on 500 real samples and an augmentation ratio of 300% allows for a median performance of 0.911 AUC, which is a 0.007 increase from the fully real baseline and only slightly less than when training on 1000 real samples (0.916 AUC). Nevertheless, the above-mentioned trend plateaus when the initial amount of real data is already substantial and the classifier performance trained solely on real data already exceeds 0.9 AUC. Particularly, when the initial amount of real data is 10,000 samples, then adding synthetic data can actually lead to a slight decrease in the median test performance (0.946 on only real data to 0.943 when the ratio is 50% and 200%).

## 4. Discussion

### 4.1. Stable Diffusion Can Be Fine-Tuned to Generate Histological Image Patches

Pretrained latent diffusion models are excellent tools for synthetic image generation, although they require fine-tuning on target images paired with textual descriptions to bridge the domain gap between the pretraining and target distributions. This was made clear by the inability of the out-of-the-box SD model to generate realistic histopathology images, which can be explained by the lack of digital pathology images in LAION, the large-scale dataset used for training vanilla SD [40]. Further supporting this, there is also a visually obvious increase in image quality after fine-tuning with in-domain data.

The class-based baseline prompt-building approach initially employed for fine-tuning SD allowed for some degree of control over the presence of cancerous or healthy features during image generation, which underlines SD’s ability to represent complex biomedical concepts when captioned data are available. However, the yielded synthetic set still lacks variability and seems to represent reduced color and morphological information when compared to the real set, which could limit their utility for clinically relevant downstream tasks. In accordance with the literature [18,54], we hypothesize that the limited image diversity that we see when using the baseline approach is due to the lack of conditioning variables in the dataset’s metadata. In turn, this limits the descriptiveness of the constructed prompts, thereby hampering SD’s ability to represent the full breadth of the real data distribution.

### 4.2. Synthetic Image Quality Can Be Improved Using Implicit Image Features

This paper presents MEP, a novel prompt-building approach to enable high-quality histopathological image generation using an open-source text-to-image diffusion model (Stable Diffusion) despite the lack of detailed textual descriptions and additional metadata. We studied the ability of the presented method to improve upon the class-conditional baseline. Quantitative and qualitative image quality assessments reveal that the synthetic image fidelity, quality, and coverage of the real distribution increases when using the MEP methodology. In particular, using MEP resulted in an 88.6 decrease in the FID score. The randomly selected synthetic image sets clearly depict an increased variety of staining and tissue morphology profiles compared to the baseline, which also better resembles the real data. These results indicate that fine-tuning an SD model leveraging additional conditional information extracted from the images themselves contributes to more realistic and diverse synthetic images, confirming the initial hypothesis.

The precision and recall metrics demonstrate a 0.11 increase in precision and a slight 0.078 decrease in recall. While this still supports the superiority of the proposed method, it is incongruent with the UMAP embeddings visualized in Figure 4. Given the UMAP embedding distributions, we would expect an improved recall and a somewhat similar precision, rather than the opposite. We hypothesize two possible explanations. First, since the recall metric is computed in the less compressed Inception-V3 feature space, it might have captured some slight loss of realistic diversity that does not translate onto the UMAP projected space. This slightly lower recall might account for a subset of less realistic features introduced by the proposed method that could not be noticeable to the human eye but are picked up by the Inception-V3 model. Alternatively, since the precision and recall computations rely on the approximation of the real and synthetic manifolds, a potentially non-uniformly dense data distribution could lead to faulty approximations [44] and hence skewed metric values.

### 4.3. Implicit Image Features Can Be Used to Generate Images That Yield Better Cancer-Detection Performance When Training on Only Synthetic Data

In another test to address synthetic generation performance, a classifier was trained solely with synthetic data (Table 2). The fact that neither of the synthetic-based performances degrade to chance values (0.5 AUC) supports the claim that the generated synthetic data are still relevant to the cancer-detection task. This test also indicates the advantage of the proposed MEP approach relative to the baseline: although training on only synthetic data is still not sufficient to reach the performance when training on only real data, the higher performance when training on images generated with the MEP method compared to the baseline method seems to confirm its improved ability to replicate the “cancer” and “healthy” tissue features present in the real dataset. Furthermore, this shows that the potential loss of realistic diversity suggested by the slightly lower recall values had no effect on the data’s downstream usability for training CAD algorithms.

### 4.4. Implicit Image Features Yield Images That Can Replace Additional Real Training Data in Low-Data Regimes

Despite some concerns regarding patch resolution, the supplementary blind synthetic image evaluation revealed the distinction between real and synthetic samples to be a challenging task even for domain experts. A detailed discussion about the strengths and limitations of said evaluation can be found in the supplementary Section B.2. These results, coupled with the previously discussed image quality assessment and synthetic-only classifier training results, show that the generated images closely resemble real ones. In turn, this suggests their potential suitability for serving as real data replacements in settings where obtaining more real training data is either too costly or limited by privacy concerns.

Investigating classification performance when training on multiple proportions of real and synthetic data generated by the proposed MEP methodology tells us several things. Performance boosts comparable to using larger amounts of real data are achieved when using synthetic data rather than real data in low-data regimes (see Figure 5). This shows that the high image quality and diversity produced by the proposed MEP approach can be translated into actual practical utility while, once again, underlining the relevance of the synthetic data to the cancer-detection task.

Notably, this experimental setting relies on a larger corpus of data for training the generator model than for training the classifier. The results then indicate that our solution could offer value to data owners by enabling the generation of synthetic data from their private datasets, even with limited annotations. This synthetic data could then be shared alongside small subsets of real private data. This way, not only would data acquisition costs be lowered but also privacy concerns related to the use of real data would be mitigated.

### 4.5. Implications, Limitations, and Future Work

Recent studies in histopathological image synthesis have explored the potential of LDMs for generating high-quality images. However, these methods have limitations, such as reliance on pixel-wise annotations, the non-availability of comprehensive descriptive information for all regions of whole-slide images (WSIs), and costly requirements for extensive and diverse datasets. In this work, the comparison of image quality between the two explored prompt-building approaches strongly suggests the value of our method for generating high-quality images from datasets with limited annotations. This is particularly relevant in histopathology as the field lacks extensive and well-annotated datasets like those of other medical domains [55,56]. In Histopathology, annotations on the WSI level are much more readily available than annotations for specific regions, patches, or pixels, as routine pathological examination reports are usually recorded on the organ or case level. Moreover, if pixel-level annotations are required, their creation usually involves non-standard, time-consuming, and costly processes. Therefore, considering the gigapixel scale of WSIs and the technical challenges associated with their processing, most datasets and AI models in this field are usually restricted to a collection of small patches with incomplete annotations, which, as shown here, limits the fidelity and realistic diversity of the images derived by these generative models. By circumventing the need for extensive annotations, our approach fills a critical gap in histopathology image generation, as it holds the potential to extend the advantages of synthetic data to a broader range of medically relevant applications where these are lacking.

Given the differences between the scale and source of the data used for LDM training, the chosen experimental setting does not allow for a fair comparison between the MEP approach and the solutions from previous works [33,34,35]. These works aim to achieve state-of-the-art performance in histopathology image synthesis, and therefore they utilize all the amount of data that they have available. Since our main objective is to evaluate the benefit of the proposed MEP method over the binary class conditional baseline, we require a different, more controlled data setting (see Section 2.4). Nevertheless, the improvement over the baseline image quality discussed above suggests that MEP could stand as a promising alternative that allows for histopathology image generation even in low-data regimes where neither pixel- nor WSI-level annotations are available. We leave for future work a comparison between the proposed solution and the remaining from the literature in the same data setting.

The particular focus of this work was a cancer-tissue-detection task. Notwithstanding, the choice of this particular downstream application was enforced merely by the dataset in use (PCam). In fact, the presented framework is not linked to a specific tissue or set of classes, thereby being equally applicable to any other medically relevant disease-classification task in the digital pathology domain that suffers from a lack of annotated data. In addition, our evaluation confirmed the hypothesis that richer and more detailed conditional information ultimately leads to better image synthesis quality. More detailed templates that allow for the addition of extra descriptive information to the prompts will certainly lead to further quality improvements. Indeed, the positive results presented here should serve as motivation for further investigations not only into different and richer templates but also into the utility of the method across diverse scenarios. Moreover, even though the MEP solution is particularly relevant in the digital pathology field due to the known constraints related to annotation availability, it could also be applied to other imaging domains where comprehensive annotations might be lacking, with minimal modifications in the text prompt template.

Nevertheless, our prompt-building approach presents some limitations that should be addressed in derivative works. Despite their efficacy, both prompt-building approaches rely on templates, which fall short in terms of prompt interpretability. This limitation arises from its reliance solely on the cluster indexes, which do not convey any explicit description of the shared image features that they represent. Adding to that, the templates lead to a rigid prompt set that might hinder the SD model’s ability to understand free-form user input in other downstream applications [57]. GPT-based models have been used to rephrase, summarize, and simplify text in a number of applications [34,58,59], with notable success in improving performance in other natural language processing tasks [60]. Introducing greater semantic variability in our prompts is left as future work.

Furthermore, achieving a test set AUC of over 0.85 with just 100 real training samples from an initial dataset of over 200,000 samples (see Figure 5) suggests that the dataset and classification task are notably straightforward [61]. Therefore, our results on this task might underestimate the benefits of synthetic data in other, more challenging tasks. We leave this experimentation for future work too.

Additionally, the PCam dataset is limited to relatively small patches. Our method produced images at a theoretical resolution of 0.1823 μm/px and an area of 93.3 μm^2^. However, given the downsampling originally performed in PCam, the overall information present in the image (i.e., the detail of visible cellular structures) approximately corresponds the information content of a scan at 10× and 0.972 μm/px. In this sense, evaluating the MEP method for the creation of larger and “higher magnification” images would be promising. As an example, synthetic images with a magnification corresponding to a “standard” 40× objective (e.g., 0.25 μm/px) or a larger field of view (e.g., 0.237 μm^2^) could be generated [62,63].

Finally, it would be interesting to explore whether using a feature extractor model that has been pretrained on histopathology images, rather than natural ones, would improve the overall performance of MEP, as using domain-specific feature extractors has been reported to outperform general purpose ones [55,64].

## 5. Conclusions

In this work, we take an important step towards histopathology patch generation with text-to-image diffusion models from datasets without comprehensive metadata. Our method proved effective at addressing that limitation by obtaining meaningful conditions in an unsupervised manner from the data itself. These conditions were shown to enhance synthetic image quality and diversity in comparison to class-conditional image generation alone. The improved real-data performance of downstream classifiers trained on synthetic data suggests that generated images may also replicate discriminative features between different classes more reliably. Despite some concerns related to the patch resolution, the blinded evaluation by expert pathologists further supported these conclusions as the distinction between real and synthetic images proved to be challenging. Additionally, we show that these synthetic images can be used jointly with small subsets of real data to minimize the amount of real data needed to train a model with limited-to-no loss in performance. All things considered, the work presented here has two main implications. First, it provides a promising solution (MEP) for extending the benefits of synthetic data to a wider range of medical applications where extensively annotated data are scarce. And second, MEP shows promising potential for reducing the data acquisition costs and privacy concerns related to training CAD systems, improving their accessibility and, consequently, their medical utility.

## 6. Ethical and Clinical Implications

The use of synthetic data in diagnostic histopathology, while promising, raises several ethical and clinical concerns. Synthetic images may not capture the full diversity of real-world samples as, just like any other machine learning algorithm, they are limited by the representativeness and quality of their training data. In the clinic, if these models are not trained nor used cautiously, they can potentially introduce biases, leading to unequal treatment outcomes. There is also a risk of diagnostic inaccuracies if synthetic data fail to replicate the nuances of actual pathology, which can compromise patient safety. Integrating synthetic data into clinical workflows requires caution to avoid skewing clinical judgment. To minimize these risks, clinicians should use synthetic data as a supplementary tool rather than the primary source for diagnostics, and it is essential to validate synthetic data against a diverse range of real-world cases to ensure representativeness. While results in the field of synthetic data generation have been promising, clinicians should be vigilant about potential biases and regularly cross-reference synthetic data with actual patient samples to maintain diagnostic integrity.

## Figures and Tables

**Figure 1 diagnostics-14-01442-f001:**
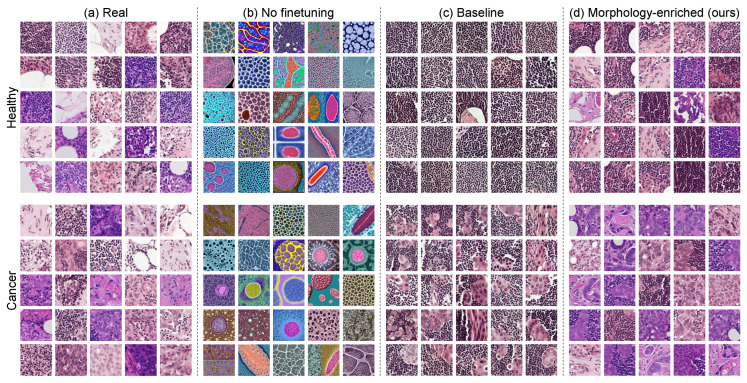
Randomly selected subset of 25 samples for (**a**) the real dataset, (**b**) the synthetic set generated by Stable Diffusion (SD) out-of-the-box without fine-tuning, (**c**) the synthetic set generated by an SD model fine-tuned on histopathology data using a naïve prompt-building approach, and (**d**) the synthetic set generated by an SD model fine-tuned on histopathology data using our proposed prompt-building approach that leverages semantic information for improved generative diversity. Image grids are categorized per label (healthy and cancer).

**Figure 2 diagnostics-14-01442-f002:**
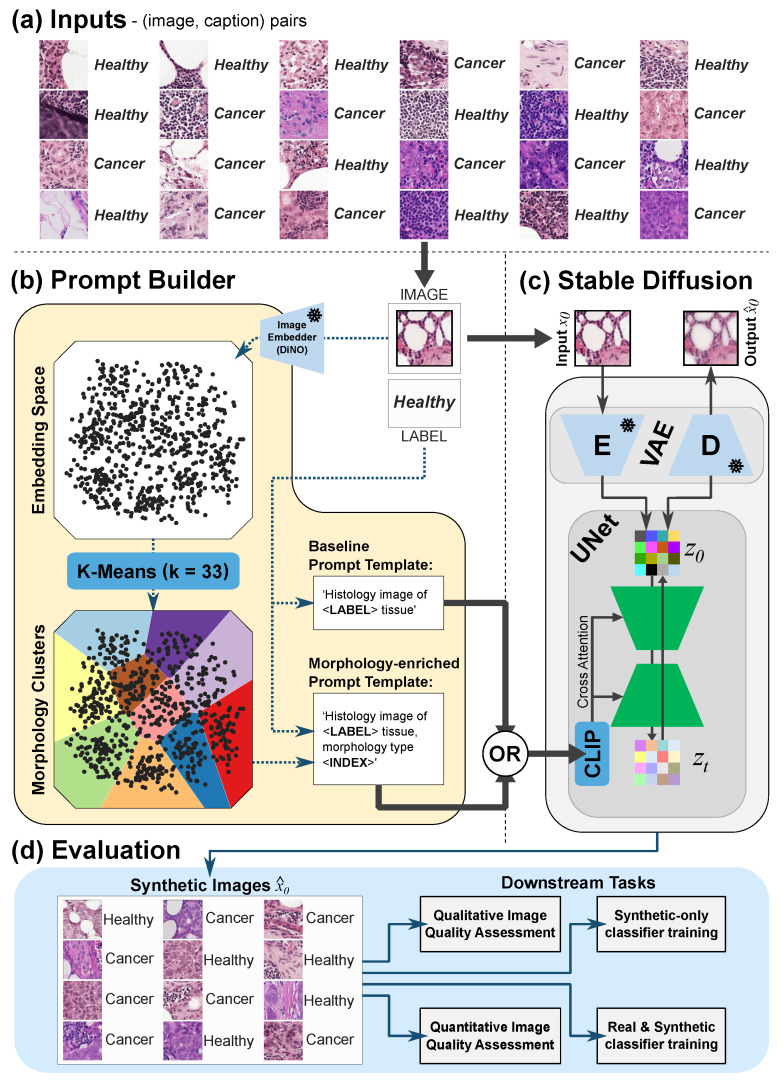
Overview of the proposed pipeline. The inputs (**a**) consist of (image and label) pairs as curated from Patch Camelyon (PCam). The prompt-building pipeline (**b**) takes said inputs to construct a prompt (or caption) that describes each image in the input dataset. Two approaches are compared: the baseline approach, in which only the label is used to generate a textual descriptor for the image, and the morphology-enriched approach, in which a frozen image embedder (DiNO [39]) is used in combination with the patch’s label to automatically extract semantic features from the image (clustered into 33 morphology types) to generate a morphology-rich prompt. After prompt building, Stable Diffusion (**c**), an open-source latent diffusion model (LDM), is trained using either of the prompt-building approaches from (**b**). Stable Diffusion is based on a variational autoencoder (VAE) and a UNet. The VAE uses its encoder (E) to reduce the dimensionality of the input image into a latent ($z_{0}$) and can recover full-resolution images using its decoder (D). The VAE’s latent ($z_{0}$) is used by the UNet, alongside the information in the prompt (via CLIP, a textual embedding model) to generate synthetic images. After model training, the performance of the fine-tuned Stable Diffusion model is evaluated on a series of downstream tasks (**d**). For this purpose, a large array of synthetic images is generated and tested using a Visual Turing test, standard image quality metrics, and two classification approaches. The snowflake icon corresponds to a frozen model.

**Figure 3 diagnostics-14-01442-f003:**
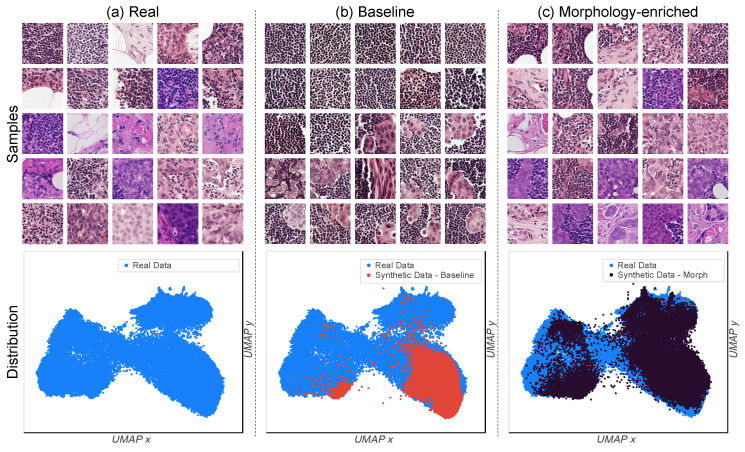
Randomly selected subset of 25 samples for the real (**a**), baseline (**b**), and morphology-enriched (**c**) datasets. The rightmost column depicts the coverage comparison of the real data distribution between the two synthetic datasets. The manifold representation is generated based on Inception-V3 latents with a 2D UMAP transform.

**Figure 4 diagnostics-14-01442-f004:**
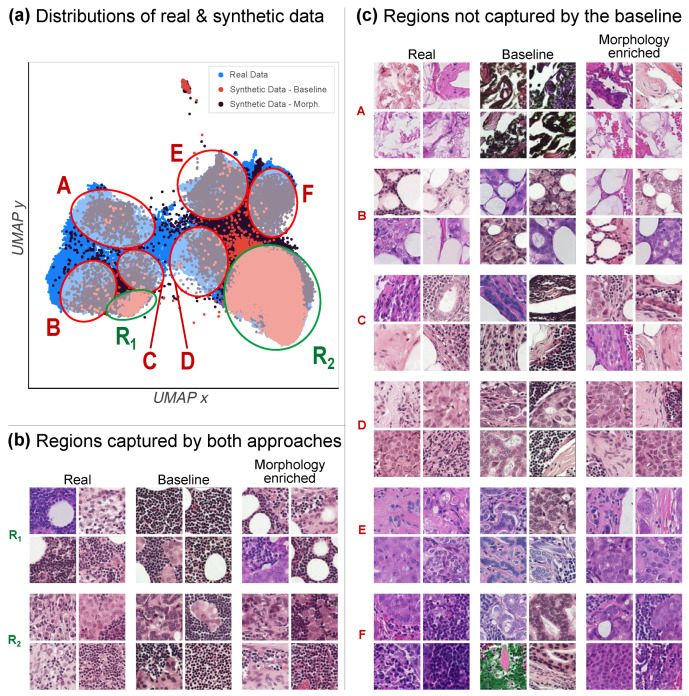
(**a**) UMAP embedding distributions: the real data distribution (blue) is better covered by the morphology-enriched prompt building (black) as compared to the baseline prompts (red). Overlaid on the figure are regions R_1_ and R_2_ (in green), which are captured by both prompt-building approaches, and for which examples are selected in (**b**). Also overlaid are regions A through F (in red), which represent regions not captured by the baseline approach but well represented by the morphology-enriched prompt building, with selected examples being depicted in (**c**). It is to be noted that the baseline approach is prone to visual outliers in regions A–F (e.g., green tincture in region F and darker images in region A). Zoom for details.

**Figure 5 diagnostics-14-01442-f005:**
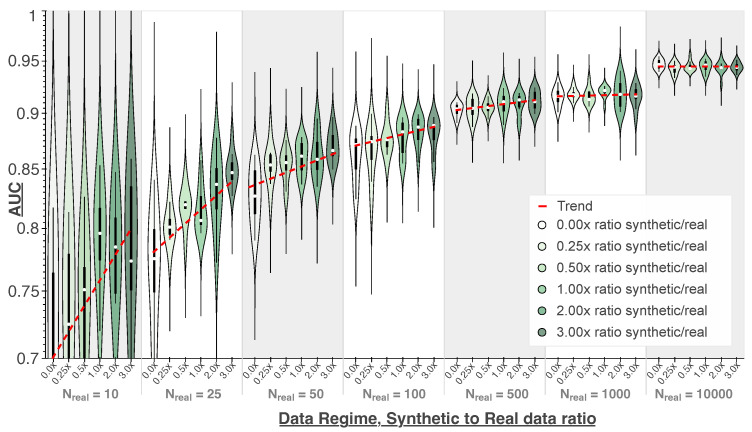
Distribution of test set AUC values across different folds for a classifier trained on varying proportions of real and synthetic data. Each group of violin plots represents a data regime with an initial real data size (N real) between 10 to 10,000 samples. For each N real, the multiple violin plots illustrate the test set AUC distribution when training the classifier with increasing amounts of synthetic data generated using our approach at augmentation ratios of 25%, 50%, 100%, 200%, and 300% relative to the real data size (i.e., augmentation ratio of 200% for data regime 100 corresponds to adding 200 synthetic samples to the 100 real samples training set). The violin plots depict both a box plot and kernel density plot, which allows for better visualization of the performance distribution of the 10 models trained for each setting. The median values are represented by a white dot.

**Table 1 diagnostics-14-01442-t001:** **Metrics comparison for the different synthetic dataset generation approaches.** Comparison of FID, improved precision, and improved recall metrics for the synthetic datasets generated via the baseline and morphology-enriched (MEP) approach. Bold values indicate best performance, arrows indicate direction of improvement in the metric.

Prompting Strategy	FID ↓	Precision ↑	Recall ↑
Baseline	178.8	0.065	**0.218**
MEP	**90.2**	**0.175**	0.140

**Table 2 diagnostics-14-01442-t002:** **Classification AUC when training with real-only or synthetic-only data.** Comparison of test set AUC when training a ResNet-34 on real data and on synthetic data generated from each of the explored prompt-building approaches. Bold values indicate best performance, arrow indicates direction of improvement in the metric.

Training Dataset	AUC ↑
Real Data	0.960
Synthetic Data—Baseline	0.773
Synthetic Data—MEP	**0.805**

## Data Availability

All data are fully available without restriction at https://www.kaggle.com/competitions/histopathologic-cancer-detection/data accessed 1 February 2023.

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
