# Peer review of "Latent Diffusion Models with Image-Derived Annotations for Enhanced AI-Assisted Cancer Diagnosis in Histopathology"

_diagnostics, 2024, doi:10.3390/diagnostics14131442_

Round 1

Reviewer 1 Report

Comments and Suggestions for Authors
  1. The quality of synthetic data generated by latent diffusion models may not fully replicate the complexity and variability of real histopathological images, potentially leading to biases or inaccuracies in AI model training.

  2. The effectiveness of constructing structured textual prompts from automatically extracted image features depends on the quality and relevance of the extracted features, which may vary.

  3. The reliance on detailed textual descriptions in common implementations of latent diffusion models may pose a challenge in domains like histopathology where such descriptions are not readily available.

  4. Pathologists found it challenging to detect synthetic images, indicating potential limitations in the visual fidelity of the generated images.

  5. The generalizability of the proposed method to other datasets with different characteristics or labeling schemes needs further validation.

  6. Ethical considerations surrounding the use of synthetic data in medical applications, particularly in diagnostic histopathology, need to be addressed to ensure patient care and outcomes are not compromised.

  7. The process of generating synthetic data heavily relies on the quality and diversity of the original dataset. If the original dataset is limited in scope or biased in some way, it may negatively impact the quality and usefulness of the synthetic data.

  8. Despite the improvement in Fréchet Inception Distance (FID) observed with the inclusion of image-derived features in the prompts, there might still be challenges in achieving consistent performance across different datasets or pathology types.

  9. The method's reliance on automatically extracted image features introduces a potential risk of information loss or distortion during feature extraction, which could impact the accuracy and relevance of the constructed textual prompts.

  10. While synthetic data may effectively train AI models, there may still be limitations in the model's ability to generalize to real-world clinical scenarios, especially when faced with novel or rare histopathological presentations not adequately represented in the training data.

  11. The interpretability of AI models trained on synthetic data may be compromised, raising concerns about the transparency and accountability of diagnostic decisions made by these models in clinical settings.

  12. The computational resources required for generating and processing synthetic data using latent diffusion models may be substantial, posing practical challenges for implementation in resource-constrained settings or low-resource healthcare environments.

Author Response

Comments 1. The quality of synthetic data generated by latent diffusion models may not fully replicate the complexity and variability of real histopathological images, potentially leading to biases or inaccuracies in AI model training.

Response 1: Thank you for pointing this out. We agree with this comment, as the data distribution captured by a latent diffusion model is just an approximation of the data distribution found in real histopathology images. Although this, as you well put, might lead to biases and inaccuracies when training AI models, LDMs are the models that are able to capture this distribution best within the literature. Still, we argue that this fact does not invalidate our results, given that the model is able to cover the distribution, as shown in the manuscript.

Comments 2. The effectiveness of constructing structured textual prompts from automatically extracted image features depends on the quality and relevance of the extracted features, which may vary.

Response 2: Absolutely. However, in this case the scientific literature finds itself in a predicament. We neither have enough textual/captioned data to be able to train such an LDM model nor we have other types of models like automatic captioners so that those captions can be generated. We agree that the quality of the captions would depend on the feature extractor, which in this case is a DiNO embedder. Although other histopathology specific embedders exist, such as Dippel et al, these high-quality embedders are not openly available to the scientific community, hindering our work. Although we agree that our methodology could benefit from better embedders, this does not change the conceptual validity of our approach and is a very interesting direction to follow in the future.

Comments 3. The reliance on detailed textual descriptions in common implementations of latent diffusion models may pose a challenge in domains like histopathology where such descriptions are not readily available.

Response 3: We agree with this. This is what our method is attempting to solve: the automatically-generated textual descriptors we use for training our LDM model allow us to improve its performance and to cover larger regions of the data manifold, as shown in Figures 3 and 4. This is why we believe our method is useful, as we are trying to improve LDM training by extending the existing, non-descriptive labels (True/False) with quantifiable (albeit improvable) information about the image in question.

Comments 4. Pathologists found it challenging to detect synthetic images, indicating potential limitations in the visual fidelity of the generated images.

Response 4: We believe this is not a limitation but an advantage of the proposed method. If the synthetic images were easily identifiable by pathologists, that would mean that the generated images are very visually different from real data. The fact that pathologies find difficulties at identifying them means that our method is able to replicate histopathology images, indicating high image quality. We refer you to section B1 of the supplementary material for a more in-depth discussion.

Comments 5. The generalizability of the proposed method to other datasets with different characteristics or labeling schemes needs further validation.

Response 5: Thank you for your comment. We believe our evaluation is a good starting point for applying LDM methodologies to datasets with lack of comprehensive annotations. Our evaluation was designed as a proof-of-concept study to assess the viability of the method. For that purpose, we chose a large and heterogenous publicly available dataset that includes images from 400 patients and two clinical centers. While there is certainly scope for further assessment of generalizability, we believe this would be best left to a further, dedicated and extensive study.

Comments 6. Ethical considerations surrounding the use of synthetic data in medical applications, particularly in diagnostic histopathology, need to be addressed to ensure patient care and outcomes are not compromised.

Response 6: We appreciate the reviewer’s concerns about the ethical use of synthetic data in diagnostic histopathology. We acknowledge that synthetic data poses several potential risks to patient care, and we have added a paragraph “Ethical and Clinical Implications” to our manuscript (Section 6). While an exhaustive discussion of ethical and clinical implications in pathology would go beyond the scope of our manuscript, we recommend the following literature for a more in-depth exploration of ethical and clinical implications:

1.      Giuffré et al., Harnessing the power of synthetic data in healthcare: innovation, application, and privacy. npj Digit. Med. (2023): This paper examines the benefits and challenges of using synthetic data in healthcare analytics, discussing its potential to enhance data privacy and predictive analytics, while addressing issues of data quality, bias, and privacy risks, and emphasizing the need for regulatory measures to ensure ethical and responsible use. (Link)

2.      Mcduff et al., Synthetic Data in Healthcare, preprint (2023): This paper discusses challenges, risks and opportunities for synthetic data, in structured and unstructured data domains in healthcare. (Link)

We believe these references provide valuable insights and further detail on the ethical and clinical considerations involved in using synthetic data in medical diagnostics.

Comments 7. The process of generating synthetic data heavily relies on the quality and diversity of the original dataset. If the original dataset is limited in scope or biased in some way, it may negatively impact the quality and usefulness of the synthetic data.

Response 7: We agree that the quality and diversity of the original dataset is essential and that synthetic data will simply reproduce existing biases if used naively. Therefore, when using our method, it is essential to assess and understand the limitations and biases in all data sources. This is true of all generative models, and indeed of any machine learning approach. We have added an extra discussion about the ethical and clinical implications of using synthetic data, where we believe to have highlighted the importance of the representativeness of the training data (see the response to Comment 6 and the new Section 6 of the manuscript).

Comments 8. Despite the improvement in Fréchet Inception Distance (FID) observed with the inclusion of image-derived features in the prompts, there might still be challenges in achieving consistent performance across different datasets or pathology types.

Response 8: Thank you for providing this comment. We view our evaluation as an initial step in the application of LDM methodologies to datasets with limited comprehensive annotations. Our assessment was designed as a proof-of-concept study aimed at evaluating the method's feasibility. To achieve this, we utilized a large and diverse publicly available dataset comprising images from 400 patients and two clinical centers. While there is room for further assessment of generalizability, we believe that this would be best addressed in a dedicated and extensive future study.

Comments 9. The method's reliance on automatically extracted image features introduces a potential risk of information loss or distortion during feature extraction, which could impact the accuracy and relevance of the constructed textual prompts.

Response 9: In this regard, we completely agree. Still, the AI literature surrounding LDM models has recently shown that image fidelity is improved both when using automatic methods to describe images and when using GPT-like models to rephrase captions to extend them. As an example, DALL-E 3 uses VLG, an automatic captioner developed on very large datasets, to provide additional context to an already captioned image. Then, DALL-E 3 also uses a GPT model to merge the captions and to provide a lengthier description of the image to be generated. This has proven to improve the alignment between the caption and the generated image. This, however, is not practically doable in histopathology, as no automatic captioners exist for said imaging domain. In this scenario, we believe that a more modest automatic feature extraction approach, as the one proposed in our work, can help bridge the very large gap between the natural image and the medical image domains.

Comments 10. While synthetic data may effectively train AI models, there may still be limitations in the model's ability to generalize to real-world clinical scenarios, especially when faced with novel or rare histopathological presentations not adequately represented in the training data.

Response 10: Thank you for your comment. The risk of the training data not encompassing novel or rare histopathological presentations is indeed a limiting factor for the model’s ability to generate realistic images of uncommon conditions. As mentioned earlier, the revised version now includes a new section where we acknowledge and discuss the need for representativeness of the training data in order safely use this technology in the clinic (Section 6). All the while, this limitation is not specific to our work but rather something that affects every histopathology-oriented predictive or generative model. Besides that, this project was not so much focused on addressing generalization to novel or rare conditions but more on demonstrating the feasibility of using image-derived features to improve synthetic image quality. That said, to adequately test our method we ensure a more controlled data setting, where introducing scarce data from rare conditions would not make sense. We believe our protocol consists of a good starting point for validating MEP and hope to dedicate some effort in future works to evaluate it further on even more diverse datasets including rare conditions.

Comments 11. The interpretability of AI models trained on synthetic data may be compromised, raising concerns about the transparency and accountability of diagnostic decisions made by these models in clinical settings.

Response 11: We agree, addressing the concern of interpretability in AI models trained on synthetic data, especially within clinical diagnostics, is crucial for ensuring trust and an open question in the scientific community. We believe that our methodology addresses this by employing a hybrid data strategy. Although some tests have been performed using synthetic data only, all applications would revolve around utilizing synthetic data to supplement, rather than replace, real data training sets (see Sections 2.6.4 and 4.4). This maintains a crucial connection to real-world data, preserving the model's interpretability.

Comments 12. The computational resources required for generating and processing synthetic data using latent diffusion models may be substantial, posing practical challenges for implementation in resource-constrained settings or low-resource healthcare environments.

Response 12: We agree this is a concern that the scientific community should face. In fact, most of the advancements in modern LDM systems such as Stable Diffusion 3 or, DALL-E 3 revolve around reducing computational complexity, in some cases even halving it. Although we agree this is a concern when using these services in healthcare systems, this technology is emerging and computational constraints can be iteratively addressed. The formulation proposed in this work is agnostic to the specific formulation of diffusion model, so a more compute-efficient or low-resource friendly version could be trained for the needs of each end-user.

4. Response to Comments on the Quality of English Language

English language fine. No issues detected

5. Additional clarifications

No additional clarifications.

Reviewer 2 Report

Comments and Suggestions for Authors

Comment for MS- 3018945

I am grateful for the opportunity to review the manuscript entitled “Latent Diffusion Models with Image-Derived Annotations for Enhanced AI-Assisted Cancer Diagnosis in Histopathology.” The MS is well-written and informative for the reader. There is a minor suggestion for me.

I want to offer some suggestions for improvement, which may help the authors to strengthen their study:

1.     Please provide the reason for model selection of the inceptionv3 model for your study.

2.     Could you please provide references for model selection of other studies about cancer histology?

3.     Could you please compare the strengths and weaknesses of your studies with other studies in the same area of interest in the discussion section?

Author Response

Comments 1. Please provide the reason for model selection of the inceptionv3 model for your study.

Response 1: Thank you for raising this point. The Fréchet inception distance (FID) metric is the standard and most common image quality metric for evaluating the similarity between real and synthetic image distributions, serving as proxy for generative model performance. Per definition, the FID score is computed over features extracted by InceptionV3 model, pretrained on a large amount of images, allowing to compare the two image sets based on denser image representations. Following a similar approach as to the authors of the FID score and other generative AI works, we used the same InceptionV3 compressed image representations as a basis for computing other distribution-based metrics like the precision and recall.

Comments 2. Could you please provide references for model selection of other studies about cancer histology?

Response 2: Thank you for your comment. We are afraid that we do not fully understand the question. Would you like to know more about other architectures for the synthetic generation of histopathology images? If so, we would point towards Mogahdam et al (arXiv:2209.13167) and from Xu et al (arXiv:2304.01053). These two works use a variety of Diffusion Denoising Probabilistic Model architectures where the denoising happens in the image space rather than in the latent space of a VAE like in the LDM architecture employed in this work. Other works, such as Xue’s (arXiv:2111.06399) and Levine’s (10.1002/path.5509) experiment with Generative Adversarial Networks. We found extending the Background section with a more in-depth analysis of these methods beyond the scope of work, as our proposed MEP methodology focus on the latest LDM architecture. Particularly, it leverages the specific text-based conditioning mechanism proposed by the Stable Diffusion team, which did not exist for these earlier architectures. If this is not what you were expecting, we would like to ask you for clarification. Thank you for your understanding.

Comments 3. Could you please compare the strengths and weaknesses of your studies with other studies in the same area of interest in the discussion section?

Response 3: Thank you for the comment. As far as we are aware, our paper is the only work revolving around training and evaluating an LDM on the PatchCamelyon dataset, which complicates comparison with the literature. We chose this dataset given its lack of comprehensive pixel-wise annotations, which allow us to show the effectiveness of our MEP methodology at improving image synthesis quality in annotation-scarce settings. That being said, there is a fundamental difference between the objective of our work and most works in the literature, as they mostly focus on trying to reach state-of-the-art histopathology image synthesis at the cost of large quantities of comprehensively annotated data which are particularly scarce in this field (see Discussion, Section 4.5, Lines 484-494). Our work’s main contribution is the proposal and validation of a technique that seems promising at circumventing this data requirement. In the Introduction we aimed at centering the description of the methods proposed for image synthesis in histopathology around this need for annotations (see Lines 36-70). Nevertheless, we agree that the Discussion might benefit from some extra contextualization, so we have complemented it with a summary of this introductory analysis (see Discussion, Section 4.5, Lines 478-484).

4. Response to Comments on the Quality of English Language

English language fine. No issues detected

5. Additional clarifications

No additional clarifications.

Reviewer 3 Report

Comments and Suggestions for Authors

In this study, the authors propose an approach that uses latent diffusion models to generate histopathology images. The article is written in a fluent manner.

My evaluations are as follows:

What criteria were used to select these prompt templates? Would the results of this approach have been improved if different templates were chosen?

How were the results tested with unknown test data after training on the training data from Kaggle? Why wasn't the training data, which has known ground truth values, divided into training, test, and validation sets?

The current contributions' impact on the literature and novelty have not been sufficiently emphasized.

Some figure captions are as long as paragraphs, reducing the readability of the article. It is recommended to shorten the figure captions.

Author Response

 Comments 1. What criteria were used to select these prompt templates? Would the results of this approach have been improved if different templates were chosen?

Response 1: This is a very interesting and relevant comment. In this regard, the prompt templates were selected in their most basic form to address the utility of this proof of concept. We purposefully strayed away from making complex prompts to control only one degree of freedom at a time. However, as you correctly state, this could have been addressed in many different ways – e.g., a segmentation algorithm could have been used to quantify information in the patches to provide more information, or other additional image embedders could have been used. In this regard, the design decisions could have widely varied, which would have changed the model’s ability to disentangle the synthetic image manifold. We have added some discussion about this in Line 514 of the Discussion, Section 4.5.

Comments 2. How were the results tested with unknown test data after training on the training data from Kaggle? Why wasn't the training data, which has known ground truth values, divided into training, test, and validation sets?

Response 2: Thank you for your question, we are happy to provide further clarity on this point. Indeed, we are using the training data from a specific Kaggle challenge, whose link you can find here. This version of the PatchCamelyon dataset comes already split into a training set, whose labels are provided, and a hidden test set, whose labels are not available. To test our models, we had to submit the predictions of every model directly to Kaggle via its API, from which we could gather a single test metric. Conducting testing on a blind set through the Kaggle API is expected to bolster the reliability of our results, as it eliminates the possibility of data leakage. For the validation set, this one was indeed split from the Kaggle training set as described in Section 2.4.

Comments 3. The current contributions' impact on the literature and novelty have not been sufficiently emphasized.

Response 3: We appreciate the comment, which also resonates with the points raised by other reviewers. We have complemented the Discussion with a summary of the limitations of previous works (analyzed at depth in the Introduction, Background paragraph, Lines 36-70), hoping that this contextualizes and highlights our contributions and their impact.

Comments 4. Some figure captions are as long as paragraphs, reducing the readability of the article. It is recommended to shorten the figure captions.

Response 4: Thank you for the recommendation. We have taken your feedback into consideration and have made slight adjustments to the captions of Figure 1 and 5. While we believe that the detailed captions are essential for accurately conveying the information in the Figures, we are open to discussing specific aspects that could be further refined. Your input is valuable, and we appreciate the opportunity to enhance the overall readability of the article.

4. Response to Comments on the Quality of English Language

English language fine. No issues detected

5. Additional clarifications

No additional clarifications.

Reviewer 4 Report

Comments and Suggestions for Authors

This paper presented a method for constructing structured textual cues from automatically extracted image features. Specifically, when using Stable Diffusion to generate pathology images, the cues are no longer limited to pure text labels, but use the Morphology Enriched Prompt-building (MEP) method proposed by the authors, which extracts morphological text descriptors from H&E image patches as an additional conditioning variable for Stable Diffusion. Compared with the baseline model, the colors of the generated images are richer than the baseline model, and the distribution of the generated images is closer to the distribution of the real data than the baseline model. And in some standard image quality metrics, all metrics except recall are improved. And in classifier training, the AUC of the images generated by the authors' method as training data is also higher than that of the images generated by the baseline model as training data. The authors also compare the distribution of AUC values of classifiers trained using different proportions of real and synthetic data in different collapsed test sets, indicating that the performance of the classification model can be well improved using this method with fewer real samples. I would like to recommend this paper for publication subject to the following comments.

1. It is suggested that some of the table notes in the text refer to a baseline that refers to Stable Diffusion without MEP.

2.A few more metrics could be added to the classifier performance comparison section for a more comprehensive comparison.

3. It is suggested that the source code be made public. Or if it is already publicly available, the website where it is located should be stated in the paper.

4. The format of Table A3 in the additional material needs to be adjusted.

Author Response

Comments 1. It is suggested that some of the table notes in the text refer to a baseline that refers to Stable Diffusion without Morphology Enriched Prompt Building (MEP).

Response 1: Thank you for your comment. We acknowledge this could have been clearer. We have adapted the caption of Tables 1 and 2 to clarify that “baseline” refers to “Stable Diffusion without Morphology-Enriched Prompt Building (MEP)”.

Comments 2. A few more metrics could be added to the classifier performance comparison section for a more comprehensive comparison.

Response 2: Thank you for your suggestion. However, the choice of only reporting the area under the curve (AUC) is a condition imposed by Kaggle based testing setting. The test set we are using is blind and, thus does not come with the corresponding ground truths. To get test results we submitted the classifier test predictions to the Kaggle API from which it is only possible to retrieve a single measure of test classification performance, the AUC metric (see Methods Section 2.1, from Line 124). Nevertheless, we believe this metric is sufficient for the classifier performance comparison, due to mainly two reasons. Firstly, unlike other metrics like sensitivity and specificity, the AUC is independent of the operating point, providing a better estimate of performance.  Secondly, likely as a result of the previous point, it is the most widely utilized metric for evaluating classifiers.

Comments 3. It is suggested that the source code be made public. Or if it is already publicly available, the website where it is located should be stated in the paper.

Response 3: Thank you for pointing this out. We apologize that the code is not yet available: we are currently in the process of obtaining the necessary open-source approvals. However, we have already added the repository link to the manuscript (at the end of the Abstract). Regrettably this repository is still private at the time of writing, but it will be publicly available before publication.

Comments 4. The format of Table A3 in the additional material needs to be adjusted.

Response 4: Thank you for bringing our attention to this. The Table A3 has been reformatted to address text overflow and caption was shortened to improve readability. All other tables were adjusted to match visual style.

4. Response to Comments on the Quality of English Language

English language fine. No issues detected

5. Additional clarifications

No additional clarifications.

Round 2

Reviewer 1 Report

Comments and Suggestions for Authors

Accept in current form

Reviewer 4 Report

Comments and Suggestions for Authors

I am satisfied with the answers the author provided to my questions.